# Effects of Elexacaftor/Tezacaftor/Ivacaftor on Cardiorespiratory Polygraphy Parameters and Respiratory Muscle Strength in Cystic Fibrosis Patients with Severe Lung Disease

**DOI:** 10.3390/genes14020449

**Published:** 2023-02-09

**Authors:** Alessandro Giallongo, Giuseppe Fabio Parisi, Maria Papale, Sara Manti, Enza Mulé, Donatella Aloisio, Vito Terlizzi, Novella Rotolo, Salvatore Leonardi

**Affiliations:** 1Pediatric Respiratory and Cystic Fibrosis Unit, Department of Clinical and Experimental Medicine, San Marco Hospital, University of Catania, 95121 Catania, Italy; 2Pediatric Unit, Department of Human and Pediatric Pathology “Gaetano Barresi”, AOUP G. Martino, University of Messina, Via Consolare Valeria, 1, 98124 Messina, Italy; 3Cystic Fibrosis Regional Reference Center, Department of Pediatric Medicine, Meyer Children’s Hospital IRCCS, 50139 Firenze, Italy

**Keywords:** cystic fibrosis, elexacaftor/tezacaftor/ivacaftor, CFTR, modulators, cardiorespiratory polygraphy

## Abstract

Background: Cystic fibrosis transmembrane conductance regulator (CFTR) modulators represent targeted therapies directly acting on the CFTR channel. The triple therapy Elexacaftor/Tezacaftor/Ivacaftor (ELX/TEZ/IVA) has been demonstrated to improve lung function and quality of life in cystic fibrosis (CF) patients. However, the effects of ELX/TEZ/IVA on sleep-disordered breathing (SDB) and respiratory muscle strength are poorly studied. The aim of this study was to assess the effects of ELX/TEZ/IVA in patients with CF and severe lung disease on cardiorespiratory polygraphy parameters, maximum inspiratory pressure (MIP) and maximum expiratory pressure (MEP) measures. Methods: patients with CF aged ≥ 12 who started treatment in a compassionate use program were retrospectively studied through the evaluation of nocturnal cardiorespiratory polygraphy parameters, MIP and MEP; and six-minute walk test (6MWT) at baseline and at months 3, 6, and 12 of treatment. Results: Nine patients (mean age 30.3 ± 6.5 years) with severe CF (mean baseline ppFEV1 34.6 ± 5.1%) were evaluated. A significant improvement in nocturnal oxygenation measured by mean SpO_2_ (92.4 vs. 96.4%, *p* < 0.05), time spent with SpO_2_ ≤ 90% (−12.6, −14.6, −15.2 min from baseline at months 3, 6, and 12, respectively, *p* < 0.05), and respiratory rate (RR) was shown, at month 12 and across the time points compared with baseline, as well as in respiratory muscle strength, although only the change in MEP was significant. Conclusions: We provide further evidence on the efficacy of the CFTR modulators ELX/TEZ/IVA, adding information about their effect on the respiratory muscles’ performance and cardiorespiratory polygraphy parameters in CF patients with severe lung disease.

## 1. Introduction

Cystic fibrosis (CF) is an autosomal recessive disorder caused by mutations in the human cystic fibrosis transmembrane conductance regulator (CFTR) gene, which is located on the long arm of chromosome 7 (7q31.2) [1].

Since the discovery of the CFTR gene in 1989, over 2000 variants have been reported, 401 of which are known to be CF-causing [1,2,3]. Specifically, mutations have been categorized into seven classes, based on the underlying defect [4]. Among European patients with CF, 87.2% were heterozygous or homozygous for a class II mutation, which determines CFTR misfolding and premature degradation, and, in almost all cases, this was F508del, thus, representing the most common CFTR mutation [5].

The European Cystic Fibrosis Society Patient Registry (ECFS) and US Cystic Fibrosis Foundation annual reports estimated that more than 80,000 people are affected by CF [6].

During the last few years, the prognosis of CF patients has notably improved due to the continuously updated protocols and treatments. In 2018, the reported median age at death in Europe was 29 years and, even more interestingly, the estimated median age of survival has increased up to nearly 50 years in some high-income countries [6,7,8,9]. Indeed, if for years, symptomatic drugs, such as pancreatic enzymes and antibiotics, have represented the only available treatment for CF [10], the recent insights into genetics, pathophysiology of CFTR dysfunction, and pharmacology have paved the way for the development of potentially more effective targeted therapies, known as CFTR modulators [11]. These are small molecules classified as potentiators and correctors, based on their mechanism of action. Ivacaftor (IVA), a CFTR potentiator, which increases the CFTR channel open probability, was the first molecule to be tested in patients carrying gating mutations in a randomized clinical trial (RCT) [12]. Subsequently, in order to extend treatment with CFTR modulators to a larger number of patients, other molecules, such as Lumacaftor/Ivacaftor (LUM/IVA), were evaluated [13]. Among these, Lumacaftor (VX-809), which acts as a CFTR corrector in CFTR protein misfolding, mainly induced by F508del, enhanced the stability of the CFTR protein, thereby increasing its expression on the cell membrane [13]. Since the F508del mutation also induces impaired gating function, the efficacy of Lumacaftor/Ivacaftor (LUM/IVA) association was assessed in an RCT. [13]. Nevertheless, CFTR correction remained variable and partial [14]. Hence, to improve the rescue of CFTR channels, it was suggested to add another corrector, Elexacaftor (VX-445) (ELX), in combination with Tezacaftor/Ivacaftor (TEZ/IVA) [15]. The triple therapy ELX/TEZ/IVA was effective in improving the forced expiratory volume in 1 s (FEV1), rate of pulmonary exacerbations, and quality of life (QoL) in patients homozygous for the F508del mutation, as well as in those with F508del minimal function (MF) mutation [16,17,18,19,20,21,22]. To date, ELX/TEZ/IVA is licensed by the European Medicines Agency to treat patients with CF aged six or older with at least one F508del mutation [23].

Currently, the effects of ELX/TEZ/IVA on sleep-disordered breathing (SDB) and respiratory muscle strength are poorly studied [24,25]. SDB is common in chronic lung disease such as CF and correlates with impaired QoL and outcome [26,27]. Cardiorespiratory polygraphy is a non-invasive tool, detecting SDB, which can occur early in the course of the disease, such as nocturnal hypoxemia, which is reported even in patients with normal daytime oxygen saturation [28,29].

The respiratory muscles can also be affected in patients with CF. Several factors have been suggested to contribute to impaired skeletal muscle function, including pulmonary hyperinflation, physical inactivity, systemic inflammation, and even CFTR channel dysfunction [30,31,32]. Maximum inspiratory pressure (MIP) and maximum expiratory pressure (MEP) generated by the respiratory muscles are two easy-to-perform measurements that allow one to estimate the strength of the respiratory muscles. Few and conflicting data on this index exist in people with CF [33]. MIP has been found to be significantly decreased in CF patients with severe disease, while MEP has been reported to be positively correlated with lung function and exercise tolerance [34,35]. Hence, respiratory muscle dysfunction, may contribute to impaired QoL and, ultimately, chronic respiratory failure by reducing exercise tolerance. The six-minute walk test (6MWT) represents a reproducible and reliable test to evaluate functional exercise capacity in CF [36]. It is a feasible and non-expensive test, showing good correlation with the prognosis [37].

The aim of this study was to assess the efficacy of treatment with the triple-combination-therapy ELX/TEZ/IVA on a cluster of patients with CF and advanced pulmonary disease, investigating the effects on cardiorespiratory polygraphy parameters, muscle strength measures, and exercise tolerance.

## 2. Materials and Methods

### 2.1. Design and Participants

A retrospective study at the Pediatric Respiratory and Cystic Fibrosis Unit, San Marco Hospital, University of Catania, from March 2020 to September 2021, including patients affected by CF who had the opportunity to start treatment with the triple combination CFTR modulating therapy ELX/TEZ/IVA for compassionate use before commercialization in Italy (July 2021). The study was conducted in accordance with the Declaration of Helsinki and written informed consent was obtained from the patients.

### 2.2. Inclusion Criteria

Patients who met both of the following criteria were included:Diagnosis of CF based on clinical presentation of CF (positive newborn screening test, suggestive signs or symptoms, and family history) plus positive sweat chloride test (Cl ≥ 60 mmol/L) or clinical presentation plus borderline sweat chloride results (Cl > 30 mmol/L and Cl < 60 mmol/L) plus two CF-causing CFTR mutations or CFTR genotype undefined or unknown based on genetic analysis plus positive CFTR physiologic testing showing CFTR dysfunction [38];The inclusion criteria for compassionate use included a genotype heterozygous for F508del and one qualifying MF allele after whole CFTR gene scanning, age > 12 years, and severe lung disease, defined as either highest percentage predicted FEV1 (ppFEV1) < 40% in the preceding 2 months or being on a lung transplant waiting list [19].

### 2.3. Dosing

Each patient received the following treatment doses of ELX/TEZ/IVA: two tablets in the morning (each containing 75 mg IVA, 50 mg TEZA, and 100 mg ELX) and one ivacaftor tablet (150 mg) in the evening.

### 2.4. Outcome Measures

The primary endpoint was to evaluate absolute change from baseline in cardiorespiratory polygraphy parameters. Secondary outcomes were absolute change in MIP and MEP, ppFEV1, and 6MWT. Measures were performed at baseline, month 3, 6, and 12 from the start of treatment.

### 2.5. Nocturnal Cardiorespiratory Polygraphy

Patients underwent overnight registration at the hospital (SomnoScreen^®^ Plus TM Domino, software v.2.3.1) on their usual respiratory support, if needed, measuring the following parameters: the apnea–hypopnea index (AHI), average number of apneas and hypopnea episodes per hour; the oxygen desaturation index (ODI), mean number of oxygen saturation falls ≥3% per hour; mean oxygen saturation (SpO_2_); cumulative time spent with SpO_2_ ≤ 90%; mean respiratory rate (RR). Specifically, apnea is defined as an episode of respiratory arrest lasting more than 10 s. Hypopnea is defined as a drop in respiratory flow ≥50% associated with a decrease in oxygen saturation ≥3% or an arousal. An AHI <5/hour is considered normal for adults [39].

### 2.6. MIP and MEP

The static respiratory pressures, MIP and MEP, correlate with respiratory muscle strength. MIP is measured starting from Residual Volume (RV), while MEP is measured starting from Total Lung Capacity (TLC) according to American Thoracic Society/European Respiratory Society (ATS/ERS) statement on muscle testing [40]. Measurements were performed with the patient seated and with a mouthpiece and a nose clip. Five or more measurements with a respiratory effort of at least 1.5 s were performed until three measurements within 20% of the highest value were obtained. The highest value was included for the analysis.

### 2.7. ppFEV1

Spirometric measurements were conducted according to the ERS guidelines [41]. ppFEV1 was calculated by comparing measured FEV1 to average FEV1 for age, sex, height, body mass, and ethnicity. Values over 80% of predicted FEV1 are considered normal.

### 2.8. 6MWT

The 6MWT was conducted in accordance with the ATS guidelines, in a closed corridor, with a flat, hard surface of 30 m. It measures the distance that a patient can quickly walk in a period of 6 min, providing information about the patient’s ability to perform daily activities [42].

### 2.9. Statistical Analysis

Statistical analysis was performed using MedCalc Software. Results are expressed as mean and standard deviation (SD). The statistical analysis was performed using non-parametric tests: the Wilcoxon test for comparison between two groups and Friedman test for the comparison between more than three groups. A *p* value lower than 0.05 was considered statistically significant.

## 3. Results

In total, nine CF patients (mean age 30.3 ± 6.5 years; M/F = 2/7) with severe lung disease (mean baseline ppFEV1 34.6 ± 5.1%) were included. All of them completed a 12-month treatment course and they are on ongoing treatment. Three patients had been previously treated with LUM/IVA and then they were shifted to ELX/TEZ/IVA because of disease progression, despite treatment. The demographics and baseline features of the included patients are summarized in Table 1.

### 3.1. Nocturnal Cardiorespiratory Polygraphy Parameters

The evaluation of the primary endpoint, the absolute change from baseline in nocturnal cardiorespiratory polygraphy parameters, showed a significant improvement in nocturnal oxygenation measured by mean SpO_2_ and cumulative time spent with SpO_2_ ≤ 90% and RR at month 12 (*p* < 0.05) and across the other time points (*p* < 0.01) (Figure 1, Table 2). The improvement in the previously mentioned parameters was already observed at the first time point (month 3) and it was sustained during the following time points.

The mean of nocturnal SpO_2_ at baseline was 92.4%, with two patients having a mean of SpO_2_ ≤ 90% at baseline that increased to 96.4% after 12 months of treatment. A slight increase in the AHI (+0.3 episodes per hour) and a reduction in the ODI were reported, although these were not statistically significant (*p* > 0.05). Furthermore, none of the patients had an AHI greater than 5, which is set as the cutoff to detect obstructive sleep apnea syndrome, neither at baseline nor during follow-up. Nocturnal cardiorespiratory polygraphy was conducted with patients on their usual respiratory support. At the beginning of the treatment, seven patients were on nocturnal ventilation, four with high-flow nasal cannula (HFNC), and three with non-invasive ventilation (NIV). Five patients (three with HFNC and two with NIV) and three patients (HFNC) were on nocturnal ventilation at month 3 and month 6, respectively. The treatment led to the suspension of ventilation in four out of seven patients at month 12 (*p* > 0.05).

### 3.2. MIP and MEP

An improvement in the indexes of muscle strength (MIP and MEP) was observed at month 12 and across the time points, although the MIP did not show significant change from baseline (16.1 mmHg; *p* > 0.05) (Figure 2). The MEP increased by, on average, 22 mmHg (+37.9%) (*p* < 0.01).

### 3.3. 6MWT

Patients were able to walk progressively longer distances compared with baseline (483 m) during the 6MWT at month 3 (+78 m) and month 12 (+138 m) (*p* < 0.01).

Table 2 summarizes the results at each time point.

## 4. Discussion

We showed that the treatment with the triple-combination-therapy ELX/TEZ/IVA in nine CF patients with severe lung disease resulted in improved patients’ nocturnal SpO_2_. The improvement was observed at month 3 and it was sustained at month 12 of treatment, regarding both the mean nocturnal SpO_2_ and the cumulative time spent with SpO_2_ ≤ 90%.

Despite CF patients’ survival increasing in recent years, in most cases, their QoL is often severely affected [43] and poor sleep quality contributes to impaired QoL in CF patients with moderate-to-severe lung disease, also affecting their neurocognitive function [44,45,46]. Sleep efficiency is influenced by oxygen saturation and other variables such as comorbidities (nocturnal cough, asthma, CFTR-related disorders) [47]. Although a consensus on the definition of hypoxia in CF is lacking [48,49,50], lower nocturnal SpO_2_ nadir and lower sleep efficiency have been observed in patients with CF, both children and adults, compared with healthy controls [51]. Indeed, a higher rate of respiratory events (AHI or respiratory disturbance index) during sleep have been reported in children with CF than in controls [51]. Furthermore, patients with moderate-to-severe CF may experience nocturnal hypoxemia, despite having normal oxygen saturation during the day [52], and even patients with normal pulmonary function test or mild-to-moderate lung disease may experience nocturnal oxygen desaturation, as evidenced in 24 children with CF, 95.8% of whom showed nocturnal desaturation episodes [53]. Hypoxemia, defined as SpO_2_ < 90% for >5% total sleep time, has been observed in 6% of pediatric clinically stable patients with CF, and this correlated with the FEV1 [50,53]. Most studies have found that nocturnal oxygen saturation is directly correlated with ppFEV1, especially in patients with lower FEV1 values (<65%) [51,54,55,56,57,58,59]. In line with this, the improvement that we have shown in nocturnal oxygen saturation was consistent with that observed in FEV1.

It has been questioned whether ELX/TEZ/IVA treatment could be beneficial even in the event of advanced pulmonary disease. In our series, seven patients out of nine had a baseline ppFVE1 below 40%, indicating advanced pulmonary disease. Nevertheless, their response to treatment was significant regarding the primary endpoint, as suggested by several authors, who assessed other outcome measures [60,61,62].

Nocturnal hypoxemia has been also associated with a deterioration in pulmonary circulation and increased oxidative stress, which may lead to inflammation and increased antimicrobial resistance [48]. Hypoxia is among the several factors involved in the induction of the nuclear factor kB (NFkB), a transcription factor upstream of the inflammatory cascade, playing a key role in the immune response. The activation of NFkB releases IL-1 that is responsible for sterile inflammation and eventually lung tissue destruction [63,64]. Moreover, sleep oxygen desaturations represent a risk factor for the development of pulmonary hypertension as well [45,65]. Indeed, hypoxemia, determined by pulmonary vasoconstriction, induces inflammation and pulmonary artery remodeling with subsequent development of pulmonary hypertension, which is associated with reduced survival [45,66]. Therefore, the treatment with ELX/TEZ/IVA may affect the disease status, by improving nocturnal oxygenation and reducing inflammation, subsequently improving the patients’ outcome and QoL.

Overnight cardiorespiratory polygraphy also showed a significant reduction in the RR. We hypothesize that the reduced RR during sleep may also contribute to improved SpO_2_, by reducing the work of breathing and the oxygen needed. This might be related to the improvement observed in respiratory muscle strength measured through MIP and MEP, which could have played a role in reducing the work of breathing and the RR during sleep. Nevertheless, only the improvement in MEP reached statistical significance. This may be explained by the measurement technique: during the MIP, the patient makes a maximal inspiratory effort with closed airways, usually starting from RV, while, during MEP measurement, the patient makes a maximal expiratory effort with closed airways, usually starting from TLC. Since CF patients experience progressive loss of lung elasticity, they are not able to start a respiratory effort from TLC, so their inspiratory efforts are shorter and, paradoxically, the patient shows a severe difficulty making a forced inspiration. On the other side, during MEP measurement, the effect of gravity on expiratory muscles, forced activation, forced spring-back of the thorax, and reduction in lung volumes lead to an expulsive push greater than during inspiration.

MEP has also been reported as a predictor of physical exercise intolerance [35]. As regards physical exercise, patients underwent 6MWT and, consistently with the observed increased respiratory muscle strength, they were able to walk longer distances at month 12 compared with baseline. Only Wark et al. had previously investigated the effect of a CFTR modulator, LUM/IVA, on 6MWT for 10 CF patients, showing an increase of 118 m at 52 weeks (*p* = 0.006) [67]. Although the 6MWT cannot replace spirometric measurements, it may represent a valuable tool to expand the assessment of QoL and prognosis in CF patients [68]. Indeed, a 6MWT < 475 m has been reported to be associated with increased risk of death or requirement for lung transplant [37]. In our series, four out of nine had baseline 6MWT lower than the 475 m cutoff and all of them were above this cutoff at month 12 of treatment. Interestingly, Gambazza et al. did not find a significant positive correlation between MEP and 6MWT, suggesting the six-minute walk test works, a variable resulting from the distance walked multiplied by BMI, as a more appropriate parameter to assess this correlation. However, the abovementioned study included CF patients with FEV1 > 40% [69].

This study has several limitations, mainly the small sample size and the lack of a control group. However, the efficacy of the treatment was confirmed on a relatively long follow-up compared to the current published studies about ELX/TEZ/IVA, with all the patients completing the 12-month treatment.

## 5. Conclusions

In summary, we provided further evidence on the efficacy of the CFTR modulators ELX/TEZ/IVA, adding information about their effect on respiratory muscle performance and nocturnal cardiorespiratory polygraphy parameters, which have not been investigated up to now, in CF patients with severe lung disease. Moreover, we found that the triple combination therapy induced an improvement in nocturnal oxygen saturation. Since nocturnal hypoxemia occurs early in CF and has been associated with inflammation, we suggest that the benefits of precocious and long-term treatment with CFTR modulators could be greater than expected, interfering with the vicious cycle of inflammation-impaired mucociliary clearance infection typical of CF [70,71].

CFTR modulators represent disease-modifying drugs that have dramatically changed the prognosis of CF. After decades of symptomatic therapies, the introduction of CFTR modulators set a new milestone in the treatment of a chronic disease such as CF. New insights into genotypes, genotype–phenotype correlation, CFTR structure, and ex vivo models may allow one to develop novel molecules and/or predict the efficacy of current modulators, leading to more tailored therapy [72,73]. Future studies should assess the long-term efficacy of CFTR modulators in CF in larger samples; the efficacy in the child population at early stages of the disease; in outcome measures related to QoL, perceived symptoms, and physical activity; and their possible application in rare CF genotypes or other diseases [73].

## Figures and Tables

**Figure 1 genes-14-00449-f001:**
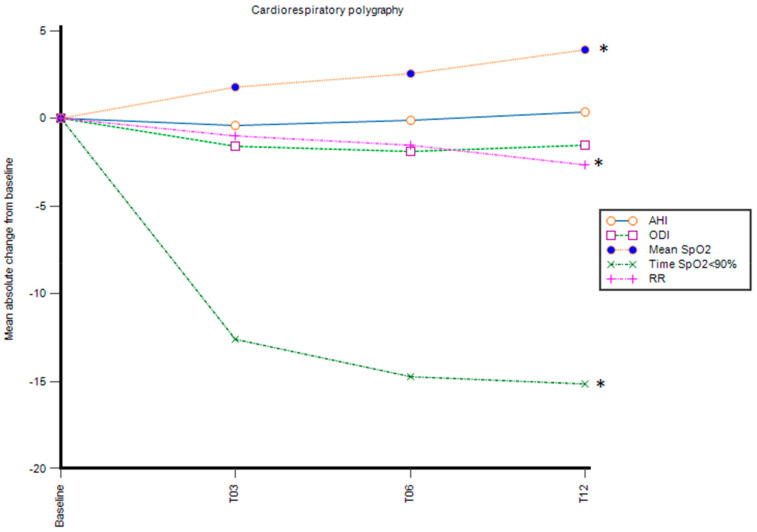
Mean of the absolute change from baseline in cardiorespiratory polygraphy parameters, AHI, ODI, mean SpO_2_, time spent with SpO_2_ < 90%, respiratory rate; T03: 3 months, T06: 6 months; T12: 12 months; * *p* < 0.05.

**Figure 2 genes-14-00449-f002:**
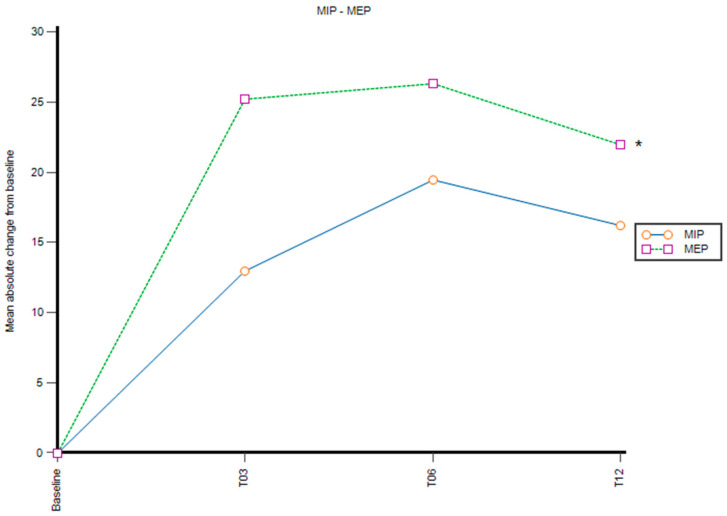
Mean of the absolute change from baseline in MIP and MEP. T03: 3 months, T06: 6 months T12: 12 months. *: *p* < 0.05.

**Table 1 genes-14-00449-t001:** Demographic and mean baseline patients’ features.

Demographics	
Patients	9
Males (%)	2 (22.2)
Age at start of treatment, mean (SD), yrs	30.3 (6.5)
Age < 18, yrs (%)	0 (0)
ppFEV1, mean (SD)	34.6 (5.1)
ppFEV1 < 40% (%)	9 (100)
*P. aeruginosa* colonization (%)	4 (44.4)
CFQ-R, mean (SD)	73.3 (17.9)
BMI, mean (SD), kg/m^2^	19.9 (2.5)
AHI, mean (SD)	1.4 (1.8)
ODI, mean (SD)	5.5 (5.8)
Mean nocturnal SpO_2_%, mean (SD)	92.4 (2.9)
time SpO_2_ ≤ 90%, min (SD)	15.7 (28.1)
RR, mean (SD)	26 (1.9)
Ventilation (%)	7 (77.8)
6MWT, mean (SD), m	483 (67.5)
MIP, mean (SD), mmHg	72.7 (32.1)
MEP, mean (SD), mmHg	58 (27.7)

Yrs, years; ppFEV1, percentage predicted forced expiratory volume in the first second; BMI, body mass index; SD, standard deviation; AHI, apnea–hypopnea index; ODI, oxygen desaturation index; 6MWT, six-minute walk test; MIP, maximum inspiratory pressure; MEP, maximum expiratory pressure.

**Table 2 genes-14-00449-t002:** Outcome measures at months 3, 6, 12, and mean absolute change from the baseline.

Outcomes	Baseline (SD)	Month 3	Month 6	Month 12	Change from Baseline	*p* Value
AHI	1.4 (1.8)	1 (0.8)	1.3 (0.8)	1.7 (0.8)	0.3	*p* > 0.05
ODI	5.5 (5.8)	3.9 (1.3)	3.8 (1.7)	4 (2.9)	−1.5	*p* > 0.05
Mean nocturnal SpO_2_	92.4 (2.9)	94.2 (1.5)	94.8 (2)	96.4 (1.6)	4	*p* = 0.03 *
time SpO_2_ ≤ 90%, min	15.7 (28.1)	3.1 (2.9)	1.1 (1.1)	0.5 (0.8)	−15.2	*p* = 0.03 *
RR	26 (1.9)	25 (2.1)	24.4 (1.4)	23.3 (0.7)	−2.7	*p* = 0.004 *
MIP, mmHg	72.7 (32.1)	85.7 (34.8)	92.1 (30.3)	88.8 (27.6)	16.1	*p* > 0.05
MEP, mmHg	58 (27.7)	83.2 (21.3)	84.3 (32.5)	80 (35.8)	22	*p* = 0.001 *
6MWT, m	483 (67.5)	513.3 (46.1)	516.7 (32.8)	573 (60.8)	90	*p* = 0.008 *

* *p* < 0.05.

## Data Availability

Data supporting the reported results are available upon request to the corresponding authors.

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
