# Peer review of "Effects of Elexacaftor/Tezacaftor/Ivacaftor on Cardiorespiratory Polygraphy Parameters and Respiratory Muscle Strength in Cystic Fibrosis Patients with Severe Lung Disease"

_genes, 2023, doi:10.3390/genes14020449_

Round 1
Reviewer 1 Report
Sleep-disordered breathing (SBD) is common in chronic lung disease like cystic fibrosis (CF) and correlates with impaired QoL and outcome. Cardiorespiratory polygraphy is a non-invasive tool, detecting sleep disordered breathing that can occur early in the course of the disease, such as nocturnal hypoxemia, which is possible reported even in patients with normal daytime oxygen saturation. Up to now the effects of ELX/TEZ/IVA on SBD and respiratory muscle strength are poorly studied. Cardiorespiratory polygraphy may be set up as a complementary evaluation within scheduled follow-up in patients with CF to early detect lung disease, thus starting early intervention, especially treatment with CFTR modulators, as well as to assess if they require ventilation. This is an interesting paper that provides a new information about the effect of the CFTR modulators ELX/TEZ/IVA on cardiorespiratory polygraphy parameters and respiratory muscle performance in CF patients with severe lung disease (ppFEV1 <40%).
As I read the text, I saw small mistakes that should be corrected or clarified.
1. Page 2, Line 70. Written - SBD is ..., shouldn't it be SDB?
2. Page 2. What is a correct spelling? "Sleep-disodered breathing'' as on Line 69 or "Sleep disodered breathing'' as on Line 72?
3. Page 3. Line 112. I think that there is no need to enter the abbreviation again, as this has already been done on page 2 on line 75. Maximum inspiratory (MIP) and expiratory pressures (MEP).
4. Page 5. Lines 159-160. It seems to me that the captions to the Figure 1 are incomplete. Where is the explanation to the drawings E and F?
5. Page 6. Line 175. This is not Figure 1, but Figure 2. Also isn't it better to label graphics as A and B for MIP and MEP?
Author Response
Sleep-disordered breathing (SBD) is common in chronic lung disease like cystic fibrosis (CF) and correlates with impaired QoL and outcome. Cardiorespiratory polygraphy is a non-invasive tool, detecting sleep disordered breathing that can occur early in the course of the disease, such as nocturnal hypoxemia, which is possible reported even in patients with normal daytime oxygen saturation. Up to now the effects of ELX/TEZ/IVA on SBD and respiratory muscle strength are poorly studied. Cardiorespiratory polygraphy may be set up as a complementary evaluation within scheduled follow-up in patients with CF to early detect lung disease, thus starting early intervention, especially treatment with CFTR modulators, as well as to assess if they require ventilation. This is an interesting paper that provides a new information about the effect of the CFTR modulators ELX/TEZ/IVA on cardiorespiratory polygraphy parameters and respiratory muscle performance in CF patients with severe lung disease (ppFEV1 <40%).
Answer: Dear Reviewer, thank you very much for the extremely positive comments that make us very proud of our work.
As I read the text, I saw small mistakes that should be corrected or clarified.
Page 2, Line 70. Written - SBD is ..., shouldn't it be SDB?
Answer: Thank you. We corrected the mistake.
Page 2. What is a correct spelling? "Sleep-disodered breathing'' as on Line 69 or "Sleep disodered breathing'' as on Line 72?
Answer: Thank you. We preferred “Sleep-disordered breathing''
Page 3. Line 112. I think that there is no need to enter the abbreviation again, as this has already been done on page 2 on line 75. Maximum inspiratory (MIP) and expiratory pressures (MEP).
Answer: Thank you. We made the correction.
Page 5. Lines 159-160. It seems to me that the captions to the Figure 1 are incomplete. Where is the explanation to the drawings E and F?
Answer: Yes, thank you. We added explanation to the drawings E and F.
Page 6. Line 175. This is not Figure 1, but Figure 2. Also isn't it better to label graphics as A and B for MIP and MEP?
Answer: Yes, thank you. We corrected the mistake.
Reviewer 2 Report
Overall this is a well written paper evaluating impact on overnight sleep study results in 9 subjects with severe CF lung disease receiving the CFTR modulator drug ELX/TEZ/IVA, and does add to our knowledge of this condition. I do think the manuscript title, abstract, and conclusions need more emphasis that the data are relevant to those with severe lung disease and not necessarily generalisable to all people with CF. I think many of the figures are unnecessary and can be removed.
Title: Needs to clarify that this study is of CF subjects with severe lung disease (line 4)
Abstract: (line 26) In general it is not useful just to report P values, without giving any values to show the clinical relevance of any change. Please give the mean SpO2 at each time-point and the percentage of time <90% SpO2, as well as the p values.
Abstract (line 30): If these subjects received ELX/TEZ/IVA through compassionate use programme, then they are likely more severely affected patients. The mean ppFEV1 of 34.6% confirms this. These results may therefore not be generalisable to all people with Cystic Fibrosis eligible for ELX/TEZ/IVA. The conclusions of the abstract need to indicate that these results are applicable to CF subjects with severe lung disease.
Introduction line 56: "set up" is not quite the right terminology, would suggest you use "evaluated".
Introduction line 70: Typo "SBD should read SDB"
Overall introduction is very long and could be made more succinct. Do we need the history of CFTR modulator development?
Results line 159. Legend for Figure 1 requires improvement. Make it clearer what the time points are (baseline, 3 months, 6 months, 12 months). Specify what the error bars are (?standard deviation). Overall I consider the number of panels excessive, all the information in Figure 1 is seen in panel A (the significant changes from baseline could be indicated by *), the other panels B-F are in my view unnecessary.
Results line 161. Please also give the mean SpO2 at month 12.
Results line 164. Clarify if all sleep studies were conducted with patients on their usual respiratory support (high flow/NIV etc), and how many were using respiratory support at each time-point (could be added to Figure 1).
Results Line 175: This should be labelled Figure 2. I don't find this figure adds to the manuscript.
Results line 179: Table 2 is a useful summary, with P values. However, only presenting change from baseline is not easy for reader to judge the clinical significance of the change. It would be better to have 2 additional columns in this table, prior to the change from baseline column, with the mean actual values of each parameter, first column at baseline, and second column at month 12. Then we can see clearly what the values changed from and to. This would also allow easier comparison of these data to other studies.
Conclusions: In general too long and not focussed. Needs to emphasise that the data presented are in CF subjects with severe lung disease, and may not be applicable to entire CF population.
In particular the lines 251-258 are not really supported by the data presented and should be removed.
References line 289: Reference 1 has "References" before first author - remove.
Author Response
Overall this is a well written paper evaluating impact on overnight sleep study results in 9 subjects with severe CF lung disease receiving the CFTR modulator drug ELX/TEZ/IVA, and does add to our knowledge of this condition. I do think the manuscript title, abstract, and conclusions need more emphasis that the data are relevant to those with severe lung disease and not necessarily generalisable to all people with CF. I think many of the figures are unnecessary and can be removed.
Title: Needs to clarify that this study is of CF subjects with severe lung disease (line 4)
Answer: Yes, thank you. According to your suggestion, we modified the title.
Abstract: (line 26) In general it is not useful just to report P values, without giving any values to show the clinical relevance of any change. Please give the mean SpO2 at each time-point and the percentage of time <90% SpO2, as well as the p values.
Answer: Yes, thank you. We added the data.
Abstract (line 30): If these subjects received ELX/TEZ/IVA through compassionate use programme, then they are likely more severely affected patients. The mean ppFEV1 of 34.6% confirms this. These results may therefore not be generalisable to all people with Cystic Fibrosis eligible for ELX/TEZ/IVA. The conclusions of the abstract need to indicate that these results are applicable to CF subjects with severe lung disease.
Answer: Yes, thank you. We specified that the results refer to people with CF and severe lung disease.
Introduction line 56: "set up" is not quite the right terminology, would suggest you use "evaluated".
Answer: Yes, thank you. We modified the sentence.
Introduction line 70: Typo "SBD should read SDB"
Answer: Yes, thank you. We corrected the typo.
Overall introduction is very long and could be made more succinct. Do we need the history of CFTR modulator development?
Answer: Dear reviewer, we have slightly reduced the introduction.
Results line 159. Legend for Figure 1 requires improvement. Make it clearer what the time points are (baseline, 3 months, 6 months, 12 months). Specify what the error bars are (?standard deviation). Overall I consider the number of panels excessive, all the information in Figure 1 is seen in panel A (the significant changes from baseline could be indicated by *), the other panels B-F are in my view unnecessary.
Answer: Dear reviewer, thank you. We revised our data and the figure 1.
Results line 161. Please also give the mean SpO2 at month 12.
Answer: Dear reviewer, thank you. We added those data.
Results line 164. Clarify if all sleep studies were conducted with patients on their usual respiratory support (high flow/NIV etc), and how many were using respiratory support at each time-point (could be added to Figure 1).
Answer: Dear reviewer, thank you. We clarified.
Results Line 175: This should be labelled Figure 2. I don't find this figure adds to the manuscript.
Answer: Yes, thank you. We corrected the typo. We would like to keep the figure.
Results line 179: Table 2 is a useful summary, with P values. However, only presenting change from baseline is not easy for reader to judge the clinical significance of the change. It would be better to have 2 additional columns in this table, prior to the change from baseline column, with the mean actual values of each parameter, first column at baseline, and second column at month 12. Then we can see clearly what the values changed from and to. This would also allow easier comparison of these data to other studies.
Answer: Yes, thank you. We revied table 2.
Conclusions: In general too long and not focussed. Needs to emphasise that the data presented are in CF subjects with severe lung disease, and may not be applicable to entire CF population.
In particular the lines 251-258 are not really supported by the data presented and should be removed.
Answer: Yes, thank you. We synthesized the conclusions.
References line 289: Reference 1 has "References" before first author - remove.
Answer: Yes, thank you. We corrected the mistake.